# Disordered Mechanical Stress and Tissue Engineering Therapies in Intervertebral Disc Degeneration

**DOI:** 10.3390/polym11071151

**Published:** 2019-07-05

**Authors:** Runze Zhao, Wanqian Liu, Tingting Xia, Li Yang

**Affiliations:** Key Laboratory of Biorheological Science and Technology, Ministry of Education, Bioengineering College, Chongqing University, Chongqing 400044, China

**Keywords:** disordered mechanical stress, intervertebral disc degeneration, tissue engineering strategies, nucleus pulposus, annulus fibrosus, limitations of therapy

## Abstract

Low back pain (LBP), commonly induced by intervertebral disc degeneration, is a lumbar disease with worldwide prevalence. However, the mechanism of degeneration remains unclear. The intervertebral disc is a nonvascular organ consisting of three components: Nucleus pulposus, annulus fibrosus, and endplate cartilages. The disc is structured to support our body motion and endure persistent external mechanical pressure. Thus, there is a close connection between force and intervertebral discs in LBP. It is well established that with aging, disordered mechanical stress profoundly influences the fate of nucleus pulposus and the alignment of collagen fibers in the annulus fibrosus. These support a new understanding that disordered mechanical stress plays an important role in the degeneration of the intervertebral discs. Tissue-engineered regenerative and reparative therapies are being developed for relieving disc degeneration and symptoms of lower back pain. In this paper, we will review the current literature available on the role of disordered mechanical stress in intervertebral disc degeneration, and evaluate the existing tissue engineering treatment strategies of the current therapies.

## 1. Introduction

Low back pain (LBP) is the most common reason for disability in developed countries accounting for 10.7% of the total population of disabled people [1]. With a three-month prevalence of as high as 40% in the United States, out of which 20%–33% of patients are unable to work, this disease has a major socio-economic impact [2]. Intervertebral disc degeneration (IDD) is a chronic disease that slowly degrades the content of intervertebral disc (IVD) and leads to unstable IVD, which limits the mobility of the spinal cord [3]. It is well established that IDD is the main contributor of LBP [4,5,6,7,8,9,10]. Depending on the stage of degeneration, current clinical strategies are divided into conservative and surgical therapies. However, these strategies are restricted to relieving the pain and symptoms without eliminating the disease itself due to an incomplete understanding of the pathobiology of IDD [11,12,13,14,15].

The main functions of the IVD are to transmit and absorb mechanical stress onto the spine and to maintain a mobile segment that allows flexion, extension, bending, and rotation [3,16]. Anatomically, a normal IVD can be roughly divided into the following three regions: (1) Nucleus pulposus (NP) with a gel-like core comprised of type II collagen and proteoglycans (Figure 1) [17,18]. NP cells are highly hydrated and account for the strength and mobility of the spine [19]. (2) Annulus fibrosus (AF) is a multi-layered fibrous tissue that surrounds the NP [20]. Unlike NP, AF is highly organized, consisting of stacked lamellae predominantly of type I collagen [21]. AF is required for transmitting stress from the NP, and maintaining the integrity of IVD and protecting it from injuries from bending, stretching, and twisting [22,23,24,25,26,27,28]. (3) End plates (EPs) are two hyaline cartilages that are enclosed, at the interfaces, by the superior and inferior vertebral bodies [29,30].

The altered biomechanics in IVD has been widely accepted as an important contributor to IDD because biomechanical loading directly affects IVD cell metabolism [31,32]. Macroscopically, overloading such as frequent bending and twisting [33], fatigue loading [34], and heavy physical work [35] raise a high risk of lumbar disc degeneration. While hypomobility, such as sedentary environment, also increase the intradiscal pressure depending on seated posture [36]. Herein, we define these aberrant altered biomechanics as disordered mechanical stress. NP stems from the axial notochord. In the embryonic stage, NP consists of notochordal cells (NCs), which gradually transform into chondrocyte-like cells (CLCs) post-embryonic and adult states [37]. A number of studies have shown that disordered mechanical stress can result in a loss of NCs, subsequently leading to IDD [38,39,40,41]. The alignment of collagen fibers is regulated by the orientation of cells in the AF, which are susceptible to mechanical stress [42,43,44,45,46,47,48]. However, disordered mechanical stress alters the structure of AF resulting from reorientation of, and that may impair the function of, AF and causes IDD. Besides NP and AF, angiogenesis is another constant feature of IDD [49] and disordered mechanical stress supported angiogenesis by inducing vascular endothelial growth factor (VEGF) [50]. Taken together, disordered mechanical stress may become another reason for IDD besides biochemistry, except for the common concept of biochemistry factor. Because disordered mechanical stress mainly induces the structure pathological changes of NP and AF, further results in the loss of mechanical function of IVD. Strategies for restoring the structures of NP and AF are essential in the clinical trials.

Evidence is accumulating on the use of tissue engineering strategies for treating IDD [51,52,53,54]. Various tissue-engineered scaffolds with feasible substances (e.g., functional cells, growth factors, proteins, peptides, etc.) are being applied in IDD therapy, achieving significant curative effects. In this paper, we will discuss the effect of disordered mechanical stress on NP, AF, and angiogenesis, and the current tissue engineering therapies in IDD. We expect that this review will support a new direction to an understanding of IDD mechanism and therapy application.

## 2. Disordered Mechanical Stress Leads to NP Degeneration

The NP is derived from the notochord during embryonic gastrulation [38,41]. During the early stages of embryogenesis, cells migrate to the endoderm transiently and form the notochordal plate that subsequently detaches from the ectoderm to form the definitive notochord, a rod-shaped embryonic structure that crosses the sclerotome. When the sclerotome starts to form the vertebral body, the notochord condenses and locates to the center of disc to form the NP [55]. NCs are the main cell type of the NP during the embryonic period. The most significant feature of NCs is that they have giant cytoplasmic vacuoles [39,56]. Studies have revealed that biosynthetic trafficking, and not endocytosis, is necessary for vacuoles formation [51].

In humans, this population of NCs is reported to decrease during the first decade of life and to have disappeared after that period [37,57]. There are, however, reports of a small percentage of cells expressing notochordal markers persisting until adult life [58]. Recent studies found that NCs secrete connective tissue growth factor that promotes the proliferation of CLCs and induces ECM secretion [59] and protects CLCs against apoptosis [60]. However, as IVD experiences long-term mechanical stress, experiments revealed that there is a negative influence of disordered mechanical stress to NCs and vacuoles. Compared with CLCs, NCs are more sensitive to mechanical stress [61]. Thus, it is notable that disordered mechanical stress-induced NCs loss may contribute to the initiation of IDD. However, other animals, such as pigs, rabbits, non-chondrodystrophic dogs, mice, and rats retain the vacuolated morphology of BCs until much later in life (Table 1) [62]. Taking into account that humans commonly lose NCs in adolescence and the cost of experimental animals (such as pigs and dogs), rodents are the favored animal model to investigate the role of NCs in IDD. In a rat tail disc degeneration model, static compression for 20 weeks significantly decreased the number of NCs and increased the expression of markers of apoptosis by day 7. Using the same compression model, researchers showed that the population of NCs a sharply decreased from 70% to less than 10% after loading [61,63]. Using genetic ablation of vacuoles or fragmentation of vacuoles in zebrafish, Ellis et al. generated larvae shorter in the anterior–posterior axis after five days post-fertilization, many of which eventually developed scoliosis of the spine later during development [64].

## 3. Disordered Mechanical Stress Leads to AF Degeneration

The AF is a fibrous connective tissue where collagen fibers align together in the same direction to form a single lamella, and adjacent lamellae are connected in opposite directions. Depending on the direction of the inner AF and the outer AF, the thickness of the collagen fiber lamellae varies between 200 and 400 μm [65]. In an adult IVD, there are about 25 lamellae circumferentially surrounding the NP [66]. Unlike NCs, cells in the AF are organized in a specific orientation. Type I collagen is the main element of the outer layers of AF, which is gradually replaced by type II collagen in the inner layers [67]. Collagen alignment is the key to the functional role of the AF. AF cells generate tractions that induce the ECM to organize along the axis of the AF cells [42,43,44,45,46,47,48]. The traction forces alter the direction of actin fibers connected to collagen fibers in the ECM [68,69] and any perturbation to actin fibers can disrupt the alignment of collagen fibers affecting the function of the AF [70]. AF cells are sensitive to mechanical stress, and mechanical function is superior to cell traction forces in collagen fibers alignment. Mechanical stress influences the AF cells through several cytoskeletal molecules such as adhesion receptors like integrin receptors, protein tyrosine kinase (PTK), and mechanosensitive channels [71,72,73,74,75,76]. A healthy IVD maintains a balance of anabolism and catabolism. Degradation of collagen fibers by zinc-dependent matrix metalloproteinases (MMPs) results in disordered stress [77]. Disordered mechanical stress induces an unbalanced loading to collagen fibers, while MMPs can degrade the unloaded or less loaded collagen fibers and destroy the architecture of AF [78,79,80]. Mechanistically, MMPs degrade collagen fibers by binding to the cleavage site. Studies have shown that disordered mechanical stress may expose the cleavage binding sites by changing the spatial structure of collagen monomers and three alpha chains [79,80,81]. Taken together, it is possible to imply that disordered mechanical stress may change the structure of IVD at cellular as well as tissue level.

## 4. Disordered Mechanical Stress Leads to Angiogenesis

Accumulating studies demonstrated that angiogenesis exists in IDD. Disordered mechanical stress facilitates the neovascularization by destroying the physical barriers, including increased lamellar disorganization and fissures [82]. Moreover, one study demonstrated that disordered mechanical stress can directly influence the ingrowth of blood vessels. Human AF cells that experienced cyclic tensile strain showed a nearly 70% increase of gene expression of pleiotrophin, in which the pleiotrophin is regarded as a pathologic alteration of disc tissue and its neovascularization [83].

In summary, there is sufficient evidence to support the concept that disordered mechanical stress can widely influence the structure and function of IVD both at the cellular and tissue level. Disordered mechanical stress induces NCs apoptosis, which promotes the proliferation of CLCs and maintains the normal function of NP [84]; the loss of NCs also means the initiation of the IDD. AF is also sensitive to disordered mechanical stress, which can change the AF structure from normal to pathological morphology, and the ingrowth of blood vessels further impairs the integrity of IVD (Figure 2). There are numerous studies that chronicle the development and application of tissue engineering-based therapies in IDD (Table 2). Although these studies are versatile, they can be divided into two areas as follows: Regenerative therapies and displacement therapies in IVD.

## 5. Tissue Engineering-Inspired Strategies to Address IDD

### 5.1. Strategies in IDD Regeneration

Based on the structure of the IVD, treatments can be divided into NP regeneration, AF regeneration, and anti-angiogenesis.

#### 5.1.1. NP Regeneration

The NP is gelatinous, consisting predominantly of type II collagen and proteoglycans, with a high water content that allows it to transmit stress and resist compressive forces when unloaded; the NP absorbs water by ionic interaction, which is ejected out into the intercellular space in response to mechanical stress [73,74,75]. In recent years, bioengineered scaffolds that are similar to the native NP structure and mechanical properties have been gaining attention. Hydrogel, which is one of the numerous biomaterials, has been pushed to the forefront of IVD treatment. Hydrogel is not biocompatible and does not have suitable mechanical properties per se. However, by changing the polymer type and optimizing the fabrication methods, hydrogel could obtain high water content, good biocompatibility, 3D network structure, and suitable biomechanical property, which is similar to the natural IVD [98]. Gan et al. created a hydrogel with dextran and gelation as the first network and poly (ethylene glycol) (PEG) as the second network to form a 3D inter-crossing network. This hydrogel was conducive to NP cell proliferation and ECM deposition in vitro, and longer NP cell retention in rat IVDs, and in the degenerated porcine IVD model, the hydrogel contributed to rehydration and regeneration of the NP [99].

Apart from being cost effective, hydrogels fabricated with high-molecular-weight polymers or native substances also perform well due to better biocompatibility, and sensitivity to the surrounding environment. Chen et al., generated a high molecular weight hyaluronic acid-gelatin-adipic acid dihydrazide (oxi-HAG-ADH) hydrogel with several advantages including: (1) Anti-inflammatory and immunosuppressive activities, necessary for clinical application; (2) low viscosity, easy to inject; (3) similar viscoelastic property as native tissue; (4) furthermore, the hydrogel supported NP cell phenotype, promoted attachment as well as proliferation, and induced the expression of type II collagen, aggrecan, Sox-9, and HIF-1A- key genes of the NP ECM [100]. With a better understanding of IDD mechanism, functions as well as NP-related proteins have been discovered; laminin, one of these functional proteins, has been demonstrated to support the attachment and biosynthesis of NP by interacting directly with NP cells [101,102,103]. Laminins are a major component of the NP ECM and interact directly with NP cells to regulate their function. Setton et al. found that laminin 111 or peptides derived from laminins can be coupled with modified PEG or poly-acrylamide gels to form functional scaffolds. These hydrogels provided a favorable environment for NP cell, proliferation, and more importantly, promoted the expression of specific markers characteristic of immature NP cells [104,105,106].

Besides laminin derived peptides, there are a number of self-assembling peptide hydrogels (SAPH) that have been used, but very few of them were evaluated for rheological behavior, a key mechanical property of NP [107,108,109]. A study adopted the FEFEFDKFK (F: Phenylalanine, E: Glutamic acid, L: Lysine) amino acid chain to form SAPH and reported no significant difference in oscillatory rheology between acellular and bovine cells-seeded on SAPH, while the total number of cells decreased with increase in load over time. Bovine cell-seeded in SAPH was less viscous and more elastic compared with the native NP. Additionally, SAPH could upregulate the expression of NP markers KRT8, KRT18, and FOXF1, and restore the NP phenotype and promote a time-dependent increase in the deposition of type II collagen and aggrecan, two crucial components of the NP ECM [110].

Safe and efficient gene therapy modalities are being widely applied as regenerative strategies in IDD. Recently, Feng et al. reported the use of high plasmid DNA (pDNA) connected to an injectable nanofibrous sponge for NP regeneration. pDNA bound to hyperbranched polymer (HP) (pDNA-HP) was complexed with modified PEG chains and allowed for self-assembly as a HP/pDNA polyplex. These polyplexes were then gathered in PLGA nanospheres and finally loaded onto nanofibrous-spongy microsphere for delivery. When injected into the NP of rat tail IDD model, this polyplex repressed fibrosis and promoted NP regeneration [111].

#### 5.1.2. AF Regeneration

The main body of the AF is composed of type I collagen, making it less hydrated and more fibrous. Transgenic mouse models, a common research tool used by scientists, have contributed to a better understanding of molecular mechanisms and cellular pathways in several human diseases including IDD. Nakamichi et al. reported that the homeobox protein Mohawk (Mkx) is crucial to the development, maintenance, and regeneration of AF. The authors found that Mkx was mainly expressed in the outer AF (OAF), and systemic ablation of Mkx in mice resulted in a deficiency of numerous tendon/ligament-related genes in the OAF, decrease in collagen fibril formation, in parallel with a rapid progression of IVD degeneration. Transplantation of mesenchymal stem cells (MSCs) overexpressing Mkx rescued the phenotype and promoted functional AF regeneration with an increase in collagen fibril formation in Mkx-/-mice [112].

In tissue engineering strategies, natural materials such as collagen, hyaluronic acid (HA), chitosan, alginate, silk fibroin, and chondroitin sulfate (CS) are well established in AF regeneration. Meanwhile, some researchers preferred to design natural biologic materials such as decellularized matrix from AF to promote tissue regeneration and repair [113,114]. Benefitting from their origins, the natural scaffolds are endowed with advantages including low toxicity, similar properties to native tissue, and easy large-scale production. Synthetic polymers are obtained from industrial products and their mechanical and physicochemical properties can be finely adjusted. The most commonly synthetic materials used for AF scaffolds include poly (trimethylene carbonate) (PTMC), poly(lactide-co-glycolide) (PLGA), poly(ε-caprolactone) (PCL), poly(D, L-lactide) (PDLLA), poly(L-lactide) (PLLA), polyurethane, and HA-poly(ethylene glycol) (PEG) [115,116]. These scaffolds can be fabricated and processed on the desired structure characteristics (aligned, angle-ply, hierarchical, bilayer, biphasic, etc.) and mechanical properties of the final engineered tissue. In a significant attempt, Pirvu et al. generated a poly (trimethylene carbonate) (PTMC) scaffold as a carrier for MSCs, which covered with a poly (ester-urethane) (PU) membrane to address AF rupture repair in a bovine IVD. In response to a dynamic load for two weeks, the composite material restored IVD height and protected the NP from herniation. MSCs implanted into the material were able to differentiate into AF cells with increased expression of type V collagen [117]. With electrospinning technique, researchers can induce the stem cells to the AF differentiated with fabricating elasticity tunable scaffolds. Zhu et al. achieved similar results on AF regeneration by growing AF-derived stem cells (AFSCs) on a biodegradable poly (ether carbonate urethane) urea (PECUU) material with comparable elasticity to native AF tissue [118].

#### 5.1.3. Anti-Angiogenesis

For suppressing the ingrowth of vessels, tissue engineering strategies normally focus on rebuilding the construction of IVD. However, accumulating evidence indicates that the degenerate NP cells act as a contributor to vessel in-growth through releasing various factors such as fibroblast growth factor (bFGF), vascular endothelial growth factor (VEGF) [119,120], and platelet-derived growth factor (PDGF) [121], as well as related pro-inflammatory cytokines including IL-1β and TNF-α. Therefore, some anti-angiogenesis hydrogels were developed against the neovascularization in IDD. One research group developed an injectable polyethylene glycol-crosslinked albumin gel (AG) that showed an angiogenic potential in IDD treatment [122]. Cell study demonstrated that endothelial cells could not adhere to the gel surface and endothelial cells showed significant lower viability compared with cells seeded on matrigel. Moreover, the AG significantly inhibited the proliferation, migration, and invasion of endothelial cells. Another research group evaluated the angiogenic potential of gellan gum (GG)-based hydrogels in NP regeneration. Their results indicated that ionic-crosslinked methacrylated GG (iGG-MA), and photo-crosslinked methacrylated GG (phGGMA) hydrogels suppressed the ingrowth of chick endothelial, while GG allowed cells infiltration, after four days of implantation [123]. A similar study used iGG-MA hydrogel containing a VEGF blocker peptidic aptamers sequence (WHLPFKC); results showed that the functional hydrogel not only prevented vessel ingrowth, but also induced their regression at the tissue/iGG–MA interface [124].

## 6. Strategies in IVD Displacement

Currently, clinical solutions to disc repair are discectomy and arthrodesis. But they are only short-term solutions for recurring low back pain or an IDD. Therefore, implant replacement has been regarded as an advanced treatment strategy to IDD. Taking into account the requirements of the clinic, implants of NP and AF should first have the similar mechanical property of native NP and AF of IVD. Second, implants may promote the survival of residential cells and anti-angiogenesis. Third, minimal/non-invasive strategies to deliver injectable materials without causing further damages to the already degenerating IVD are necessary for clinic. Due to that, numerous studies have measured the mechanical properties of native NP and AF, which is really helpful for the implants fabricating (Table 3). Taking into account that NP is rich in negative charge to maintain its water content and that loss of fixed charge results in reduced hydration and loss of disc height, synthesis of fixed charge glycosaminoglycan analogs based on sulphonate-containing polymers has proven beneficial in IVD displacement. Fixed charge implants injected into degenerated NP showed better mechanical strength when tested in vitro, and in vivo, the implant could maintain tissue hydration closer to the native NP [125]. A fabricated photo-polymerizable poly (ethylene glycol) dimethacrylate nano-fibrillated cellulose composite hydrogel that could gel in situ administered via a customized minimally invasive medical device (Figure 3) restored the function and height of degenerated IVD in bovine disc model compared to other gels, and was mechanically resistant even after half a million loading cycles. It is clear that hydrogels with advanced capabilities display a promising future for NP replacement [126]. Besides these under-developed biomaterials, a few NP implants have been extensively used for IDD treatment. The prosthetic disc nucleus (PDN) is a hydrogel that can absorb up to 80% of its weight in water. PDN has passed FDA guidelines of cytotoxicity and biomechanical tests [127]. PDN can endure up to 50 million cycles with loads ranging from 200 N to 800 N. Aquarelle is made of a semihydrated poly vinyl alcohol (PVA) hydrogel. Animal tests showed that Aquarelle has good biocompatibility and can tolerate up to 40 million cycles but high rates of extrusion were reported, ranging 20%–30% depending on the approach [128].

Instead of simply focusing on the mechanical properties, more clinical studies are required to focus on the reparative effects of these bio-materials. A bi-phasic polyurethane scaffold, consisting of a core material and a flexible electrospun envelope, was delivered into a bovine IVD model under a two-week dynamic loading cycle. Besides displaying sufficient mechanical properties during the dynamic loading, the scaffold down-regulated the expression of catabolic genes and type I collagen and stimulated proteoglycan and type II collagen deposition. Additionally, the scaffold could be customized according to the requirement of individuals [136]. Recent trials on AF displacement selected collagen/collagen-based elements as native materials for AF implantation. An 18-week study found that high-density collagen cross-linked material suppressed IDD progression. Histological examination revealed that the material was fully effective in sealing the gap rapidly and partly in joining the disrupted lamella in the disrupted AF [137]. Another study used porcine-derived pericardium as multi-laminate AF repair patches (AFRPs) where the pericardium was decellularized and assembled to generate AFRPs. When cross-linked with carbodiimide, which is resistant to collagenase, AFRPs protected the integrity and stabilized the balance of enzymatic metabolism of the degenerated IVD by promoting the migration and proliferation of AF cells in a bovine caudal IVD [97]. Sealing the gap in the disrupted AF is the main need in clinical AF treatment strategy; thus, it is necessary to evaluate the existing repair strategies. Long et al. showed that a fibrin-genipin hydrogel showed better performance due to its relatively lower risk of herniation and failure compared with other scaffold-based materials-tested. Moreover, fibrin-genipin hydrogel was the easiest to synthesize, which indicated a promising AF repair therapy [138]. To date, artificial scaffolds are difficult to satisfy the requirements of AF engineering in clinical trials because the AF tissue has a complicated structure and unevenly distributed components. With the development of decellularization technique, many researchers have shifted their attentions to the decellularized tissue ECM. ECM scaffold regulates cell survival, proliferation, and differentiation; moreover, it is an ideal carrier for growth factors and cytokines attaching and delivering in vivo. One research group developed a decellularized porcine AF scaffold by using chemical reagents and biological enzymes to remove pig AF cells. With biological and mechanical tests, results showed that the decellularized porcine AF scaffold maintained the similar structure and components compared to the native AF tissue. The mechanical property showed no significant difference between the scaffold and native AF. Most importantly, rabbit AF cells seeded into the scaffold showed good viability, implying the scaffold possessed favorable biocompatibility [113].

Whole-tissue engineering IVD combines two approaches of NP replacement and AF repair together. The methods for constructing whole IVD can be divided into the following three categories: (1) Cells-seeded scaffolds of NP and AF were prepared separately and assembled together into composite constructs. Nesti et al. used MSCs seeded PLLA electrospinning scaffold and HA gel and assembled them into an engineering IVD [139]. This composite scaffold provided a development of chondrocytic phenotype of the seeded cells. (2) Integrated biphasic NP–AF scaffolds. One research group developed an integrated biphasic NP–AF scaffolds from collagen and GAGs. A collagen–GAG co-precipitate core was comprised as the NP tissue and it is encapsulated in multiple lamellae of photo-chemically crosslinked collagen membranes, which comprise the AF-like lamellae [140]. This scaffold showed similar mechanical properties to native discs, with 82%–89% recovery of heights after mechanical loading, compared with a 99% recovery of native discs. (3) Scaffolds made of decellularized natural IVD. With chemistry and physics methods, Chan et al. made a 70% cell-removing scaffold in bovine IVDs [141]. This acellular scaffold maintains GAG content, the structure of collagen fibers, and biomechanical properties. Moreover, NP cells survive more than seven days after being implanted into the decellularized scaffold.

## 7. Concluding Remarks

In this review, we investigated the role of disordered mechanical stress in NP, AF, and angiogenesis in IDD, and current trends in tissue engineering therapies in IDD (Figure 4). It is well known that the mechanical environment can affect cellular homeostasis [142]. NP cells with a decline in osmotic pressure exhibit a decreased synthesis of aggrecan and an increased production of MMP-3, which initiate NP degeneration [21,143]. With the degeneration of the NP, increasing shear stress and decreasing swelling pressure will lead to a formation of a fibrous tissue because of the depositing of collagen type I [144]. In AF, the reduction of intradiscal pressure will lead to an inward bulging of AF, which can destroy the structure of laminae, increasing the risk of tears [145]. Thus, disordered mechanical stress plays an important role in IDD. We evaluated only a few of these therapies and concur that there are many that we have not included and are beyond the scope of this review. There were more ideas and strategies, which are beyond the scope of this article. Besides paying attention to the NP/AF repair, some studies have focused on the other factors induced by IDD. For example, pro-inflammatory cytokines such as IL-1and TNF-α, which are secreted during IDD, also promote degeneration of IVD. This study synthesized Epoxyeicosatrienoic acids (EETs) through cytochrome P450 enzymes acted on arachidonic acid. The EFTs showed a potential of being anti-inflammatory and anti-catabolism. Therefore, they were evaluated in IDD, and experiments found that EFTs could promote NP cells against apoptosis and suppress the process of IDD [146]. In clinics, tissue engineering methods are potentially available during the onset of IDD and the end stage of IDD. For example, in NP regeneration, growth factor is only sufficient to the resident cells that still respond to GF treatment. If NP cells no longer respond to the GF, a cell-seed scaffold is possible to support NP regeneration. Advanced degeneration of IVD may have a harsh environment, thus, the genetic modified cell-seeded scaffold may be employed to enhance the synthesis of GFs and allow sustained secretion of anabolic proteins [147].

With the development of tissue engineering, scaffolds that satisfy the requirements of clinical treatment are regarded as the ‘holy grail’ to IDD repair. For NP repair: (1) Implanted biomaterials need to restore the height of IVD and the motion segment stability [148,149]. (2) Implanted biomaterials should have sufficient durability, which means biomaterials can maintain physical support over millions of cycles of loading without generating minimal wear debris that may stimulate an immune response. (3) Implanted biomaterials should have a feasible environment for NP cells surviving and prevent the ingrowth of blood vessels because neovascularization facilitates the infiltration of macrophages into the IVD, triggering inflammation [150]. (4) Injectable materials are more appropriate because of its ability to cause minimal damage to the AF tissue. For AF repair: (1) Engineered AF scaffold that mimics the collagen fiber architecture of native tissue is the first choice in clinical trials. (2) Scaffolds that reproduce the mechanical properties, strength, and oriented microstructure of the native AF tissue are considered to be an ideal method for AF repair [114]. Although tissue engineering therapies are promising and versatile, their application to IDD remains challenging, in part, due to: (1) Lack of in-depth understanding of the molecular mechanisms involved in IDD, including the mechanism that regulates the transformation of NCs to CLCs. Understanding the IDD process will support guidelines for the development of compatible bioengineered materials or cells that could inhibit the loss of NCs in IDD. (2) More advanced materials are required in clinical strategies. Technologies must be developed to fabricate materials that not only can provide mechanical strength to withstand overload, but also provide a sufficient environment to promote cell proliferation and diffusion of functional proteins. (3) Lack of pre-clinical animal models for investigating degeneration and treatment. Currently, the rat tail IDD model is the most commonly used model for IDD due to easy accessibility and low cost. The bovine IVD model, on the other hand, is suitable only for in vitro studies, which can make extrapolation to in vivo difficult. Interestingly, a study reported that sheep may be a suitable model, as the IVD of sheep is an age-dependent degeneration, and due to the morphological similarity to human IVD [151].

## Figures and Tables

**Figure 1 polymers-11-01151-f001:**
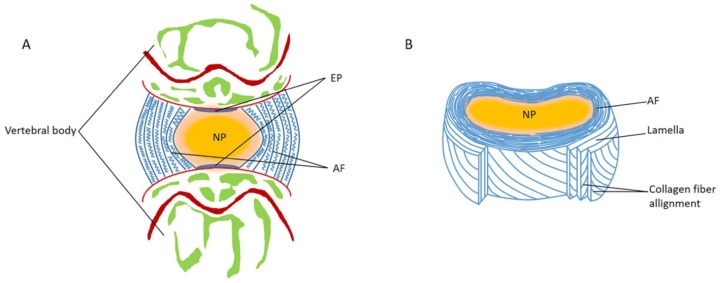
Schematic representations of the adult intervertebral disc (IVD). (**A**) Midsagittal cross-section showing anatomical regions. (**B**) Three-dimensional view of the annulus fibrosus (AF) lamellar structure.

**Figure 2 polymers-11-01151-f002:**
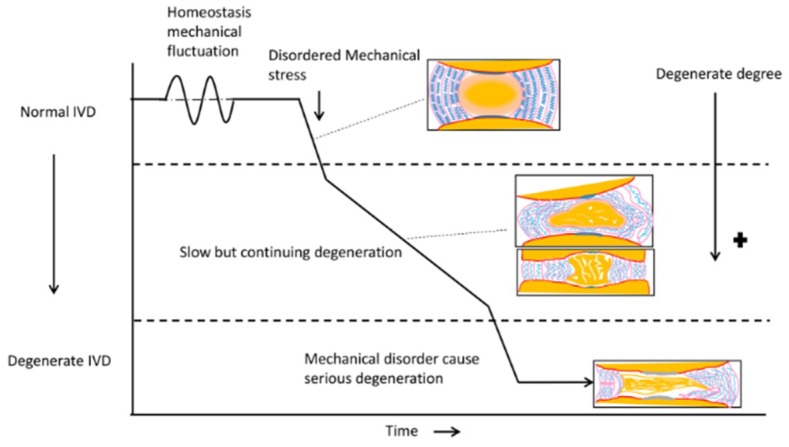
Disordered mechanical stress causes IVD degeneration. There are some mechanical fluctuations in normal IVD. Imbalance in the normal homeostatic mechanics due to mechanical disorder initiates IVD degeneration. Early stages of IVD degeneration is represented by a bulging nucleus pulposus (NP) and discrete AF. Slow but continuing degeneration results in white consolidated fibers in the NP from.

**Figure 3 polymers-11-01151-f003:**
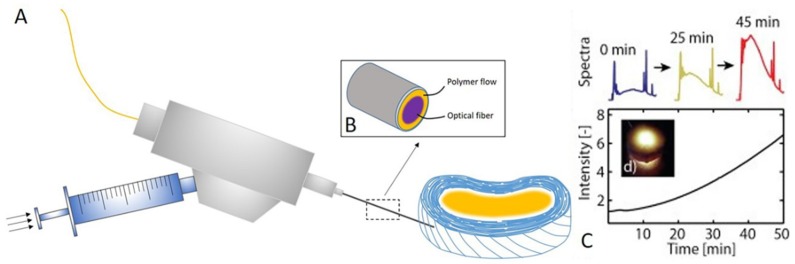
(**A**) Customized minimally invasive surgical probe used for hydrogel implantation. Liquid hydrogel precursor injection and illumination are performed by a single needle cannula. Different joints ensure high pressurization during the injection. An optical fiber permits the light delivery for photopolymerization. (**B**) Representation of the distal tip of the instrument, the hydrogel precursor flows between cannula wall, and optical fiber into the IVD. (**C**) By recording the reflected illumination spectra, signal intensity is calculated. The intensity of the signal correlates with the amount of photopolymerized material and, therefore, provides valuable information on the reaction state of the implant in real time.

**Figure 4 polymers-11-01151-f004:**
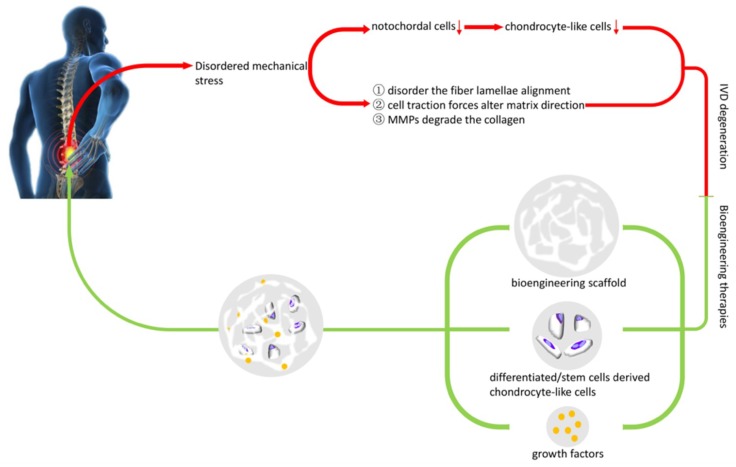
The role of disordered mechanical stress in IDD, and current developments in tissue-engineering therapies.

**Table 1 polymers-11-01151-t001:** Summary of notochordal cells indifferent species.

Species	Age of Skeletal Maturity	Age at Loss of Notochordal Cells (According to Literature)
Dog (c)	12 months	12 months
Dog (n/c)	12 months	60 months
Rabbit	10 months	6 months
Pig	12 months	Unknown
Cat	24 months	Never
Ferret	n/d	Never
Sheep	12 months	Unknown
Rat	2 months	12 months
Mouse	4 months	n/d
Human	20 years	6–10 years

c, chondrodystrophoid (beagles); n/c, non-chondrodystrophoid (mongrels); n/d: no data available. This table was cited from Christopher J. Hunter et al. 2004 [39].

**Table 2 polymers-11-01151-t002:** Summary of tissue engineering strategies in intervertebral disc degeneration (IDD) treatment.

Tissue Engineering Strategies in NP Treatment			
Materials	Test Species	Test Time	Results
PLGA	Dog	8-week	PLGA with cells significantly maintained the height and the stability of disc [85].
Fibrin	Pig	12-week	Fibrin significantly inhibited the fibrosis and inflammation of NP and enhanced the synthesis of ECM [86].
Collagen II (CII)/hyaluronate (HyA)/chondroitin-6-sulfate (6-CS)	Rabbit	84-day	The CII/HyA-CS scaffolds have a highly porous structure, high water-binding capacity, and significantly improved mechanical stability. This scaffolds also showed satisfactory biocompatibility [87].
PGA-hyaluronan	Rabbit/Sheep	12 month/6 month	Enhanced repair tissue formation and MRI intensity [88,89]
Silk fibroin (SK) /polyurethane (PU) composite	Pig	NA	SK/PU is an injectable hydrogel with minimally invasive treatment, suitable physical-mechanical properties, and visible CT and T2-weight MRI [90].
Modified hyaluronic acid gels	Pig	6-week	Both HYAFF^®^ 120 and HYADD 3^®^ treatment supported an NP-like region forming and prevented IVD narrowing, fibrous tissue replacement, and bony end-plates disruption [91].
Tissue engineering strategies in NP treatment in AF treatment			
Electrospun PCL	Rat	4-week	PCL can mimic the hierarchical organization of the native AF and achieve functional partly with native tissue [92].
Photochemically crosslinked collagen in shape of needle	Rabbit	1 month	Materials can sustain the physiologically relevant loadings, prevent leakage, and reduce osteophyte formation [93].
Collagen-fibrin gel scaffolds	Rabbit cells in vitro	4 months	Collagen-fibrin gel significantly delayed the fibrous tissue infiltration. GAG and hydroxyproline content increase over four months [94].
Tissue engineering strategies in the whole IVD			
AF-polyglycolic acid and polylactic acid NP-alginate	Mice	12-week	The engineered disc maintained the gross morphology and the AF was rich type I collagen but NP contained type II collagen [95].
AF-contracted collagen, NP-alginate	Rat	6 months	Tissue-engineered IVD maintained disc space height, produced de novo extracellular matrix, and integrated into the spine, yielding an intact motion segment with dynamic mechanical properties similar to that of native IVD [96].
AF- poly (butylene succinate-co-terephthalate) copolyester (PBST), NP-chitosan hydrogel	Rabbit	4-week	The whole TE-IVD stimulated the natural structure of IVD and retained the height of IVD after four weeks of implant [97].

**Table 3 polymers-11-01151-t003:** Summary of the mechanical properties of native AF and NP tissue.

Tissue Scale	Benchmark	Testing Methods	Mechanical Value
AF (Sub-lamella)	E	Nanoindentation	0.6–1.2 MPa
AF (Single Lamella)	E (f = 0°)	Uniaxial tension	80–120 MPa
	E (f = 90°)		0.22 MPa
AF (Multiple Lamellae)	E*_θ_* (toe/linear)	Uniaxial tension	2.5/18–45 MPa
	Axial fixed E (toe/linear)	Biaxial tension	9.8/27.2 MPa
NP	P *swell*	Confined compression	0.138 MPa
	ǀG *ǀ	Torsional shear	7.4–19.8 kPa

E = modulus; *θ* indicates the loading axes along the disc circumferential direction; toe/linear = toe-region/linear region of stress–strain curve; P *swell* = swelling pressure. This table was cited from Lewis et al. [129], Holzapfel et al. [130], Guerin and Elliott [131], O’Connell et al. [132], Johannessen and Elliott [133], and Iatridis et al. [134,135].

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
