# Peer review of "Disordered Mechanical Stress and Tissue Engineering Therapies in Intervertebral Disc Degeneration"

_polymers, 2019, doi:10.3390/polym11071151_

Round 1
Reviewer 1 Report
I think the review is well-organized, and pertinent.
However, I think the discussion has ignored mechanics - while mechanics are discussed in general terms, I think addition of some in vivo / ex vivo forces during degeneration in humans, and linking to cellular & matrix biology in humans (not just mice) would be important. I think freely mixing rodent and human IVD data is problematic, as the biology and mechanics are very different.
Therefore, I think some numbers / charts focusing on the species in which the data arise would be important. As an example, discussing notochordal cells as the central problem in nucleus in humans is problematic - we all lose them by 6-10 years of age, yet IVD degeneration patterns arise in humans 35-65 years later. The forces seen in rodents are different than that in humans.
In addition, perhaps further discussion to marry clinical conditions with proposed bioengineering treatments would be helpful. For example, in what situation would one potentially use AF engineered treatments? I can think of IVD herniations.
Or, is IVD different in degenerative scoliosis / spondylolisthesis? Which of the strategies described may be useful there?
Author Response
Replies to Reviewer 1
Comments to the Author
I think the review is well-organized, and pertinent. However, I think the discussion has ignored mechanics - while mechanics are discussed in general terms, I think addition of some in vivo / ex vivo forces during degeneration in humans, and linking to cellular & matrix biology in humans (not just mice) would be important.
Answer: Thank you for your advice. Yes, we agree with the point you addressed. So we have added some content to the concluding remarks part in the revised manuscript.
It is well known that mechanical environment can affect cellular homeostasis [1]. NP cells with a decline in osmotic pressure exhibit a decrease synthesis of aggrecan and an increase production of MMP-3 which initiate NP degeneration [2, 3]. With degeneration of the NP, an increasing shear stress and a decreasing swelling pressure will lead to a formation of a fibrous tissue because of the depositing of collagen type I [4]. In AF, the reduction of intradiscal pressure will lead to an inward bulging of AF which can destroy the structure of laminae, increasing the risk of tears [5]. Thus, disordered mechanical stress plays an important role in IDD. (Page 9, lines 344-351, red font)
Vergroesen, P.P., et al., Mechanics and biology in intervertebral disc degeneration: a vicious circle. Osteoarthritis Cartilage, 2015. 23(7): p. 1057-70.
Ishihara, H., et al., Proteoglycan synthesis in the intervertebral disk nucleus: the role of extracellular osmolality. Am J Physiol, 1997. 272(5 Pt 1): p. C1499-506.
Neidlinger-Wilke, C., et al., Interactions of environmental conditions and mechanical loads have influence on matrix turnover by nucleus pulposus cells. Journal of Orthopaedic Research, 2012. 30(1): p. 112-121.
Smith, R.L., D.R. Carter, and D.J. Schurman, Pressure and shear differentially alter human articular chondrocyte metabolism: a review. Clin Orthop Relat Res, 2004(427 Suppl): p. S89-95.
Hwang, D., et al., Role of load history in intervertebral disc mechanics and intradiscal pressure generation. Biomech Model Mechanobiol, 2012. 11(1-2): p. 95-106.
I think freely mixing rodent and human IVD data is problematic, as the biology and mechanics are very different. Therefore, I think some numbers / charts focusing on the species in which the data arise would be important. As an example, discussing notochordal cells as the central problem in nucleus in humans is problematic - we all lose them by 6-10 years of age, yet IVD degeneration patterns arise in humans 35-65 years later. The forces seen in rodents are different than that in humans.
Answer: Thank you for your advice. In this part, we mainly focus on showing the disordered mechanical stress induced a loss of NCs, which is the onset of IDD in human. According to your comments, we have added more details describing the differences of NCs between species and a table was added.
In humans, this population of NCs is reported to decrease during the first decade of life and to have disappeared after that period [1,2]. There are, however, reports of a small percentage of cells expressing notochordal markers persisting until adult life [3]. (Page 3, lines 83-85, red font)
Colombier, P., et al., Intervertebral disc regeneration: a great challenge for tissue engineers. Trends In Biotechnology, 2014. 32(9): p. 433-435.
Hunter, C.J., J.R. Matyas, and N.A. Duncan, The notochordal cell in the nucleus pulposus: a review in the context of tissue engineering. Tissue Eng, 2003. 9(4): p. 667-77.
Stosiek, P., M. Kasper, and U. Karsten, Expression of cytokeratin and vimentin in nucleus pulposus cells. Differentiation, 1988. 39(1): p. 78-81.
However, other animals, such as pigs, rabbits, non-chondrodystrophic dogs, mice and rat retain the vacuolated morphology of BCs until much later in life (Table 1.) [1]. Taking into account that human commonly lost NCs in adolescents and the cost of experimental animals (such as pigs and dogs), rodents are the favor animal model to investigate the role of NCs in IDD. (Page 3, lines 91-94, red font)
Table 1 Summary of notochordal cells indifferent species.
Species | Age of skeletal maturity | Age at loss of notochordal cells (according to literature) |
Dog (c) Dog (n/c) Rabbit Pig Cat Ferret Sheep Rat Mouse Human | 12 months 12 months 10 months 12 months 24 months n/d 12 months 2 months 4 months 20 years | 12 months 60 months 6 months Unknown Never Never Unknown 12 months n/d 6-10 years |
c, chondrodystrophoid (beagles); n/c, non-chondrodystrophoid (mongrels); n/d: no data available. This data was cited from Christopher J. Hunter et al. 2004 [2]
Miyazaki, T., et al., A phenotypic comparison of proteoglycan production of intervertebral disc cells isolated from rats, rabbits, and bovine tails; which animal model is most suitable to study tissue engineering and biological repair of human disc disorders? Tissue Eng Part A, 2009. 15(12): p. 3835-46.
Hunter, C.J., J.R. Matyas, and N.A. Duncan, Cytomorphology of notochordal and chondrocytic cells from the nucleus pulposus: a species comparison. J Anat, 2004. 205(5): p. 357-62.
In addition, perhaps further discussion to marry clinical conditions with proposed bioengineering treatments would be helpful. For example, in what situation would one potentially use AF engineered treatments? I can think of IVD herniations. Or, is IVD different in degenerative scoliosis / spondylolisthesis? Which of the strategies described may be useful there?
Answer: Thank you for your comments. According to your comment, we have collected many relative papers but as far as we know, surgical treatment is the first choice for scoliosis or spondylolisthesis. We also added some content of tissue engineering strategies in clinical IDD treatment in the revised manuscript.
In clinical, tissue engineering methods are potentially available during the onset of IDD and the end stage of IDD. For example, in NP regeneration, growth factor (GF) are only sufficient to the resident cells that still response to GF treatment. If NP cells no longer respond to the GF, a cell-seed scaffold is possible to support NP regeneration. Advanced degeneration of IVD may have a harsh environment, thus, genetic modified cell-seeded scaffold may be employed to enhance the synthesis of GFs and allow sustained secretion of anabolic proteins [1]. (Page 9, lines 359-365, red font)
O'Halloran, D.M. and A.S. Pandit, Tissue-engineering approach to regenerating the intervertebral disc. Tissue Eng, 2007. 13(8): p. 1927-54.

Reviewer 2 Report
This is a reviewing article that summarizes the mechanism of degeneration of the intervertebral disc. The author also reviews the current literature about the possible mechanism that may relieve discomfort from the degenerated intervertebral disc.
Generally, the whole manuscript is well prepared and the logic is easy to follow-up. There are few spelling errors requires the authors' attention.
Author Response
Replies to Reviewer 2
Comments to the Author
This is a reviewing article that summarizes the mechanism of degeneration of the intervertebral disc. The author also reviews the current literature about the possible mechanism that may relieve discomfort from the degenerated intervertebral disc.
Generally, the whole manuscript is well prepared and the logic is easy to follow-up. There are few spelling errors requires the authors' attention.
Answer:Thank you for your positive comment. We have corrected the spelling errors in the manuscript.
Reviewer 3 Report
Reviewer Comments
This review paper is interesting, but for general readers who are not familiar with IVD and IDD, it would help to define a few key properties that must be realized to produce the best replacements. Also, the paper would benefit from an extra review for general English grammar. It’s a good start, and will be suitable for publication after some improvements.
Content suggestions
1. Can you list a few key parameters that should be tested for engineered replacements? Maybe a list that includes rheology (including desired ranges of key criteria), ability to be sterilized, etc. That would help summarize things for a non-expert reader. Even listing the mechanical properties of the native IVD components in a table would be helpful.
2. A brief mention of which (if any) materials have made it into clinical trials or received FDA approval would be helpful. It would help the general reader to know what is the current state-of-the-art for humans.
Major edits needed
1. In the “AF regeneration,” please clarify the use of “Author et al” with proper citations and names.
Suggested edits to improve readability
1. Suggested edit: Low back pain (LBP) is the most common reason for disability in the developed countries, accounting for 10.7% of the total population of disabled people [1]. , and With a three-month prevalence up to of as high as 40% in the United States out of which 20-33% of patients are unable to work, this disease has a major socio-economic impact [2].
2. The acronym IDD is used before it is defined. Please fix.
3. Suggested edit: IDD is a chronic disease which slowly degrades the content of the intervertebral disc (IVD) and leads to unstable IVD which limits the mobility of the spinal cord [3].
4. End of line 31: add a period.
5. Line 34: due to an incomplete understanding
6. Please add an example of what can cause disordered mechanical stress.
7. Suggested edit: In rodents, the number of NCs is maintained throughout the adult life, and studies have shown that these species have delayed IDD [31-34] compared to humans because in adults NCs are replaced by CLCs
8. Suggested edit: researches showed that the population of NCs sharply decrease in the number of NC cells from 70% to less than 10% after loading
9. Suggested edit: Using genetic ablation of vacuoles or fragmentation of vacuoles in zebrafish, Ellis et al generated larvae…
10. Suggested edit: Mechanical stress influences the AF cells through several cytoskeletal molecules such as adhesion receptors like integrin receptors, protein tyrosine kinase (PTK)
11. Please rephase this confusing sentence on MMPs: It is interesting that MMPs can play as a sweeper degrade unloaded or less loaded fibers during mechanical stress [70-72].
12. Suggested edit: AF is also sensitive to disordered mechanical stress which can change the AF structure
Author Response
Replies to Reviewer 3
Comments to the Author
Reviewer Comments
This review paper is interesting, but for general readers who are not familiar with IVD and IDD, it would help to define a few key properties that must be realized to produce the best replacements. Also, the paper would benefit from an extra review for general English grammar. It’s a good start, and will be suitable for publication after some improvements.
Answer: Thank you for your kind comments on our article.
Content suggestions
Can you list a few key parameters that should be tested for engineered replacements? Maybe a list that includes rheology (including desired ranges of key criteria), ability to be sterilized, etc. That would help summarize things for a non-expert reader. Even listing the mechanical properties of the native IVD components in a table would be helpful.
Answer: Thank you for your constructive comment. We have added some content of clinical requirements of engineered replacements and a table of the mechanical properties of the native IVD component and engineered replacements.
Currently, clinical solutions to disc repair are discectomy and arthrodesis. But they are only short-term solutions to recurring low back pain or an IDD. Therefore, implants replacement has been regarded as an advanced treatment strategy to IDD. Taking into account the requirements of clinic, implants of NP and AF should first have the similar mechanical property of native NP and AF of IVD. Second, implants may promote the survival of residential cells and anti-angiogenesis. Third, minimal/non-invasive strategies to deliver injectable materials without causing further damages to the already degenerating IVD are necessary in clinic. Due to that, numerous studies have measured the mechanical properties of native NP and AF, which is really helpful for the implants fabricating (Table 2). (Page 7, lines 254-262, red font)
Table 2 Summary of the mechanical properties of native AF and NP tissue
Tissue scale | Benchmark | Testing methods | Mechanical value |
AF (Sub-lamella)
| E | Nanoindentation | 0.6–1.2 MPa |
AF (Single Lamella) | E (f=0o) | Uniaxial tension | 80–120 MPa |
E (f=90o) | 0.22 MPa | ||
AF (Multiple Lamellae) | Eθ (toe/linear) | Uniaxial tension | 2.5/18–45 MPa |
Axial fixed E (toe/linear) | Biaxial tension | 9.8/27.2MPa | |
NP | P swell | Confined compression | 0.138 MPa |
ǀG*ǀ | Torsional shear | 7.4–19.8 kPa |
E = modulus; θ indicates the loading axes along the disc circumferential direction; toe/linear = toe-region/linear region of stress–strain curve; P swell = swelling pressure. This table was cited from Lewis et al. [1], Holzapfel et al. [2], Guerin and Elliott [3], O’Connell et al. [4], Johannessen and Elliott [5] and Iatridis et al. [6,7] (Page 7, lines 280-284, red font)
Lewis, N.T., M.A. Hussain, and J.J. Mao, Investigation of nano-mechanical properties of annulus fibrosus using atomic force microscopy. Micron, 2008. 39(7): p. 1008-19.
Holzapfel, G.A., et al., Single lamellar mechanics of the human lumbar anulus fibrosus. Biomech Model Mechanobiol, 2005. 3(3): p. 125-40.
Guerin, H.L. and D.M. Elliott, Ouantifying the contributions of structure to annulus fibrosus mechanical function using a nonlinear, anisotropic, hyperelastic model. Journal of Orthopaedic Research, 2007. 25(4): p. 508-516.
O'Connell, G., et al. Biaxial Mechanics are Inhomogeneous and Altered with Degeneration in the Human Annulus Fibrosus. in Transactions of the 56rd Annual Meeting of the Orthopaedic Research Society, New Orleans, LA. 2010.
Johannessen, W. and D.M. Elliott, Effects of degeneration on the biphasic material properties of human nucleus pulposus in confined compression. Spine (Phila Pa 1976), 2005. 30(24): p. E724-9.
Iatridis, J.C., et al., The viscoelastic behavior of the non-degenerate human lumbar nucleus pulposus in shear. J Biomech, 1997. 30(10): p. 1005-13.
Nerurkar, N.L., D.M. Elliott, and R.L. Mauck, Mechanical design criteria for intervertebral disc tissue engineering. J Biomech, 2010. 43(6): p. 1017-30.
A brief mention of which (if any) materials have made it into clinical trials or received FDA approval would be helpful. It would help the general reader to know what is the current state-of-the-art for humans.
Answer: Thank you for your comments. We have added two representative NP repair materials which have passed FDA and widely applied in clinical treatment.
Besides these under developing biomaterials, a few of NP implants have been extensively used for IDD treatment. The prosthetic disc nucleus (PDN) is a hydrogel that can absorb up to 80% of its weight in water. PDN has passed FDA guidelines of cytotoxicity and biomechanical tests [1]. PDN can endure up to 50 million cycles with loads ranging from 200 N to 800 N. Aquarelle is made of a semihydrated poly vinyl alcohol (PVA) hydrogel. Animal tests showed that Aquarelle has a good biocompatibility and can tolerate up to 40 million cycles but high rates of extrusion was reported, ranging from 20%-30% depending on the approach [2]. (Page 7, lines 272-278, red font)
Ray, C.D., The PDN prosthetic disc-nucleus device. Eur Spine J, 2002. 11 Suppl 2: p. S137-42.
Allen, M.J., et al., Preclinical evaluation of a poly (vinyl alcohol) hydrogel implant as a replacement for the nucleus pulposus. Spine, 2004. 29(5): p. 515-523.
Major edits needed
In the “AF regeneration,” please clarify the use of “Author et al” with proper citations and names.
Answer: We apologize about our carelessness. In the revised manuscript, we have corrected this error.
Nakamichi R et al reported the homeobox protein Mohawk (Mkx) is crucial to the development, maintenance and regeneration of AF. (Page 6, lines 207-208, red font)
Pirvu T et al generated a poly (trimethylene carbonate) (PTMC) scaffold as a carrier for MSCs which covered with a poly (ester-urethane) (PU) membrane to address AF rupture repair in a bovine IVD. (Page 6, lines 225-227, red font)
Suggested edits to improve readability
Suggested edit: Low back pain (LBP) is the most common reason for disability in the developed countries, accounting for 10.7% of the total population of disabled people [1]. and With a three-month prevalence up to of as high as 40% in the United States out of which 20-33% of patients are unable to work, this disease has a major socio-economic impact [2].
Answer: Thank you for your comment. We have corrected this sentence.
Low back pain (LBP) is the most common reason for disability in developed countries accounting for 10.7% of the total population of disabled people [1]. With a three-month prevalence of 40% in the United States out of which 20-33% of patients are unable to work, this disease has a major socio-economic impact [2]. (Page 1, lines 26-29, red font)
Vos, T., et al., Years lived with disability (YLDs) for 1160 sequelae of 289 diseases and injuries 1990-2010: a systematic analysis for the Global Burden of Disease Study 2010. Lancet, 2012. 380(9859): p. 2163-96.
Guyer, R.D., et al., Early failure of metal-on-metal artificial disc prostheses associated with lymphocytic reaction: diagnosis and treatment experience in four cases. Spine (Phila Pa 1976), 2011. 36(7): p. E492-7.
The acronym IDD is used before it is defined. Please fix.
Answer: We apologize about our carelessness. This error has been revised in the manuscript.
Intervertebral disc degeneration (IDD) is a chronic disease which slowly degrades the content of intervertebral disc (IVD) and leads to unstable IVD which limits the mobility of the spinal cord [1]. (Page 1, lines 29-31, red font)
Risbud, M.V. and I.M. Shapiro, Role of cytokines in intervertebral disc degeneration: pain and disc content. Nat Rev Rheumatol, 2014. 10(1): p. 44-56.
Suggested edit: IDD is a chronic disease which slowly degrades the content of the intervertebral disc (IVD) and leads to unstable IVD which limits the mobility of the spinal cord [3].
Answer: Thank you for your comment. We have corrected this sentence.
Intervertebral disc degeneration (IDD) is a chronic disease which slowly degrades the content of intervertebral disc (IVD) and leads to unstable IVD which limits the mobility of the spinal cord [1]. (Page 1, lines 29-31, red font)
Risbud, M.V. and I.M. Shapiro, Role of cytokines in intervertebral disc degeneration: pain and disc content. Nat Rev Rheumatol, 2014. 10(1): p. 44-56.
End of line 31: add a period.
Answer: Thank you for your comment. A period has been added at the end of this sentence.
It is well established that IDD is the main contributor of LBP [4-10]. (Page 1, lines 31-32, red font)
Line 34: due to an incomplete understanding
Answer: Thank you for your comment. We have corrected this in the revised manuscript.
However, these strategies are restricted to relieving the pain and symptoms without eliminating the disease itself due to an incomplete understanding of the patho-biology of IDD [11-15]. (Page 1, lines 33-35, red font)
Please add an example of what can cause disordered mechanical stress.
Answer: Thank you for your comment. We have added some examples in the revised manuscript.
The altered biomechanics in IVD have been widely accepted as an important contributor to IDD because biomechanical loading directly affect IVD cell metabolism [1, 2]. Macroscopically, overloading such as frequent bending and twisting [3], fatigue loading [4], heavy physical work [5] rise a high risk of lumbar disc degeneration. While hypomobility such as sedentary environment, also increase the intradiscal pressure depending on seated posture [6]. Herein, we define these aberrant altered biomechanics as disordered mechanical stress. (Page 2, lines 49-54, red font)
Iatridis, J.C., et al., Effects of mechanical loading on intervertebral disc metabolism in vivo. J Bone Joint Surg Am, 2006. 88 Suppl 2: p. 41-6.
Wuertz, K., et al., In vivo remodeling of intervertebral discs in response to short- and long-term dynamic compression. J Orthop Res, 2009. 27(9): p. 1235-42.
Farfan, H.F., The torsional injury of the lumbar spine. Spine (Phila Pa 1976), 1984. 9(1): p. 53.
Adams, M.A. and W.C. Hutton, The effect of fatigue on the lumbar intervertebral disc. J Bone Joint Surg Br, 1983. 65(2): p. 199-203.
Rauck, R.L., et al., Chronic low back pain: new perspectives and treatment guidelines for primary care: Part II. Manag Care Interface, 1998. 11(3): p. 71-5.
Nachemson, A.L., Disc pressure measurements. Spine (Phila Pa 1976), 1981. 6(1): p. 93-7.
Suggested edit: In rodents, the number of NCs is maintained throughout the adult life, and studies have shown that these species have delayed IDD [31-34] compared to humans because in adults NCs are replaced by CLCs
Answer: Thank you for your comment. This sentence has been deleted in the revised manuscript because we added some new similar content.
Suggested edit: researches showed that the population of NCs sharply decrease in the number of NC cells from 70% to less than 10% after loading
Answer: Thank you for your comment. This sentence has been corrected in the revised manuscript.
researches showed that the population of NCs sharply decreased from 70% to less than 10% after loading (Page 3, lines 96-97, red font)
Suggested edit: Using genetic ablation of vacuoles or fragmentation of vacuoles in zebrafish, Ellis et al generated larvae…
Answer: Thank you for your comment. We have correct this sentence in the revised manuscript.
Using genetic ablation of vacuoles or fragmentation of vacuoles in zebrafish, Ellis et al, generated larvae shorter in the anterior-posterior axis after 5 days post-fertilization, many of which eventually developed scoliosis of the spine later during development (Page 3, lines 97-100, red font)
Suggested edit: Mechanical stress influences the AF cells through several cytoskeletal molecules such as adhesion receptors like integrin receptors, protein tyrosine kinase (PTK)
Answer: Thank you for your comment. We have correct this sentence in the revised manuscript.
Mechanical stress influences the AF cells through several cytoskeletal molecules such as adhesion receptors like integrin receptors, protein tyrosine kinase (PTK) and mechanosensitive channels (Page 3, lines 117-119, red font)
Please rephrase this confusing sentence on MMPs: It is interesting that MMPs can play as a sweeper degrade unloaded or less loaded fibers during mechanical stress [70-72].
Answer: Thank you for your comment. We have rephrased this sentence as ‘Disordered mechanical stress induces an unbalanced loading to collagen fibers, while MMPs can degrade the unloaded or less loaded collagen fibers and destroy the architecture of AF [79-81].’ (Page 4, lines 121-123, red font)
Suggested edit: AF is also sensitive to disordered mechanical stress which can change the AF structure
Answer: Thank you for your comment. We have revised this sentence.
AF is also sensitive to disordered mechanical stress which can change the AF structure from normal to pathological morphology… (Page 4, lines 140-142, red font)

Reviewer 4 Report
The review was focused on bioengineering strategies designed to treat disordered mechanical stress in IVD degeneration. The topic is quite interesting in the field but the title is not coherent with the content, because bioengineering comprises also permanent devices, instead the review reports just Tissue engineering strategies. However, even considering just TE solutions, the review was not well organized, some aspects were not considered, few examples were reported and just superficially discussed.
Some inputs:
-Title: should be changed accordingly with the content;
- Disordered mechanical stress leads to NP degeneration: it was reported the embryo development of NP and then some experimental studies performed on animal models (mainly rats).
- Disordered mechanical stress leads to AF degeneration: in this section a different approach was used. It was described the structure of AF in the adult and, shortly, the mechanism of degeneration. It was not taken into consideration the control of angiogenesis, that is an important factor in the functional regeneration. A summary at the end of the paragraph is not really useful.
Figure 2: the scheme proposed is really interesting and well done;
- Bioengineering-inspired strategies to address IDD: once again it was reported just strategies in the TE field;
- It was not reported strategies for the complete IVD regeneration, but NP and AF.
- Requirements of ideal structure should be reported, in particular the need of cell proliferation control and angiogenesis
- NP regeneration: Hydrogels are not biocompatible and with suitable mechanical properties per se’, but it depends on polymer/s used and fabrication methods, please revise;
- Samples must be sterile always, that processing aspect is not just an advantage;
- Page 5 lines 163-171: it were reported in vitro evaluations without mentioning the cell type used;
- AF regeneration: Just few examples were reported;
- Strategies in IVD displacement: just few examples were reported.
Author Response
Replies to Reviewer 4
The review was focused on bioengineering strategies designed to treat disordered mechanical stress in IVD degeneration. The topic is quite interesting in the field but the title is not coherent with the content, because bioengineering comprises also permanent devices, instead the review reports just Tissue engineering strategies. However, even considering just TE solutions, the review was not well organized, some aspects were not considered, few examples were reported and just superficially discussed.
Answer: We are grateful for your helpful advices. Yes, we agree with the points you addressed. So we change the title as ‘Disordered mechanical stress and tissue engineering therapies in intervertebral disc degeneration’ which is more coherent with the content. In this review, we mainly focus on the effect of disordered mechanical stress on NP and AF degeneration and the potential tissue engineering strategies which effectively restore the mechanical properties of the components of IVD. In the revised manuscript we considered more aspects especially the effect of disordered mechanical stress on angiogenesis and engineered scaffolds applications in anti-angiogenesis. More examples about AF regeneration and IVD replacement were added.
Title: should be changed accordingly with the content;
Answer: Thank you for your comment. We have changed the title as ‘Disordered mechanical stress and tissue engineering therapies in intervertebral disc degeneration’. (Page 1, lines 2-3, red font)
Disordered mechanical stress leads to NP degeneration: it was reported the embryo development of NP and then some experimental studies performed on animal models (mainly rats).
Answer: Thank you for your comment. We have added some contents to explain the reason of experimental studies prefer using rat models.
In humans, this population of NCs is reported to decrease during the first decade of life and to have disappeared after that period [1,2]. There are, however, reports of a small percentage of cells expressing notochordal markers persisting until adult life [3]. (Page 3, lines 83-85, red font)
Colombier, P., et al., Intervertebral disc regeneration: a great challenge for tissue engineers. Trends In Biotechnology, 2014. 32(9): p. 433-435.
Hunter, C.J., J.R. Matyas, and N.A. Duncan, The notochordal cell in the nucleus pulposus: a review in the context of tissue engineering. Tissue Eng, 2003. 9(4): p. 667-77.
Stosiek, P., M. Kasper, and U. Karsten, Expression of cytokeratin and vimentin in nucleus pulposus cells. Differentiation, 1988. 39(1): p. 78-81.
However, other animals, such as pigs, rabbits, non-chondrodystrophic dogs, mice and rat retain the vacuolated morphology of BCs until much later in life (Table 1.) [1]. Taking into account that human commonly lost NCs in adolescents and the cost of experimental animals (such as pigs and dogs), rodents are the favor animal model to investigate the role of NCs in IDD. (Page 3, lines 91-94, red font)
Table 1 Summary of notochordal cells indifferent species.
Species | Age of skeletal maturity | Age at loss of notochordal cells (according to literature) |
Dog (c) Dog (n/c) Rabbit Pig Cat Ferret Sheep Rat Mouse Human | 12 months 12 months 10 months 12 months 24 months n/d 12 months 2 months 4 months 20 years | 12 months 60 months 6 months Unknown Never Never Unknown 12 months n/d 6-10 years |
c, chondrodystrophoid (beagles); n/c, non-chondrodystrophoid (mongrels); n/d: no data available. This data was cited from Christopher J. Hunter et al. 2004 [2]
Miyazaki, T., et al., A phenotypic comparison of proteoglycan production of intervertebral disc cells isolated from rats, rabbits, and bovine tails; which animal model is most suitable to study tissue engineering and biological repair of human disc disorders? Tissue Eng Part A, 2009. 15(12): p. 3835-46.
Hunter, C.J., J.R. Matyas, and N.A. Duncan, Cytomorphology of notochordal and chondrocytic cells from the nucleus pulposus: a species comparison. J Anat, 2004. 205(5): p. 357-62.
Disordered mechanical stress leads to AF degeneration: in this section a different approach was used. It was described the structure of AF in the adult and, shortly, the mechanism of degeneration. It was not taken into consideration the control of angiogenesis, that is an important factor in the functional regeneration. A summary at the end of the paragraph is not really useful.
Answer: We are grateful for your helpful advice. We have added some contents of angiogenesis in the revised manuscript.
Disordered mechanical stress leads to angiogenesis
Accumulating studies demonstrated that angiogenesis was exist in IDD. Disordered mechanical stress facilitate the neovascularization through destroying the physical barriers, including increased lamellar disorganization and fissures [1]. Moreover, one study demonstrated that disordered mechanical stress can directly influence the ingrowth of blood vessel. Human AF cells experienced cyclic tensile strain showed a nearly 70% increase of gene expression of pleiotrophin, which the pleiotrophin is regarded as a pathologic alteration of disc tissue and its neovascularization [2]. (Page 4, lines 129-135, red font)
Mirza, S.K. and A.A. White, 3rd, Anatomy of intervertebral disc and pathophysiology of herniated disc disease. J Clin Laser Med Surg, 1995. 13(3): p. 131-42.
Neidlinger-Wilke, C., et al., Mechanical stimulation alters pleiotrophin and aggrecan expression by human intervertebral disc cells and influences their capacity to stimulate endothelial migration. Spine (Phila Pa 1976), 2009. 34(7): p. 663-9.
Anti-angiogenesis
For suppressing the ingrowth of vessel, normally tissue engineering strategies focus on rebuilding the construction of IVD. However, accumulating evidence indicates that the degenerate NP cells act as a contributor to vessel ingrowth through releasing various factors such as fibroblast growth factor (bFGF), vascular endothelial growth factor (VEGF) [1, 2] and platelet-derived growth factor (PDGF) [3], as well as related pro-inflammatory cytokines including IL-1β and TNF-α. Therefore, some anti-angiogenesis hydrogels were developed to against the neovascularization in IDD. One research group developed an injectable polyethylene glycol-crosslinked albumin gel (AG) that showed an angiogenic potential in IDD treatment [4]. Cell study demonstrated that endothelial cells could not adhere to the gel surface and endothelial cells showed a significant lower viability compared with cells seeded on matrigel. Moreover, the AG significantly inhibited the proliferation, migration and invasion of endothelial cells. Another research group evaluated the angiogenic potential of gellan gum (GG) based hydrogels in NP regeneration. Their results indicated that ionic-crosslinked methacrylated GG (iGG-MA), and photo-crosslinked methacrylated GG (phGGMA) hydrogels suppressed the ingrowth of chick endothelial, while GG allowed cells infiltration, after 4 days of implantation [5]. Similar study used iGG-MA hydrogel containing a VEGF blocker peptidic aptamers sequence (WHLPFKC), results showed that the functional hydrogel not only prevented vessel ingrowth, but also induce their regression at the tissue/iGG-MA interface [6]. (Page 6, lines 235-252, red font)
Fujita, N., et al., Vascular endothelial growth factor-A is a survival factor for nucleus pulposus cells in the intervertebral disc. Biochemical and Biophysical Research Communications, 2008. 372(2): p. 367-372.
Salo, J., et al., Expression of vascular endothelial growth factor receptors coincide with blood vessel in-growth and reactive bone remodelling in experimental intervertebral disc degeneration. Clin Exp Rheumatol, 2008. 26(6): p. 1018-26.
Tolonen, J., et al., Platelet-derived growth factor and vascular endothelial growth factor expression in disc herniation tissue: and immunohistochemical study. Eur Spine J, 1997. 6(1): p. 63-9.
Scholz, B., et al., Suppression of adverse angiogenesis in an albumin-based hydrogel for articular cartilage and intervertebral disc regeneration. Eur Cell Mater, 2010. 20: p. 24-36; discussion 36-7.
Silva-Correia, J., et al., Angiogenic potential of gellan-gum-based hydrogels for application in nucleus pulposus regeneration: in vivo study. Tissue Eng Part A, 2012. 18(11-12): p. 1203-12.
Perugini, V., et al., Anti-angiogenic potential of VEGF blocker dendron loaded on to gellan gum hydrogels for tissue engineering applications. J Tissue Eng Regen Med, 2018. 12(2): p. e669-e678.
Figure 2: the scheme proposed is really interesting and well done;
Answer: Thank you for your positive comment.
Bioengineering-inspired strategies to address IDD: once again it was reported just strategies in the TE field;
Answer: Thank you for your comment. We have edited it as tissue engineering-inspired strategies to address IDD. (Page 5, lines 154, red font)
It was not reported strategies for the complete IVD regeneration, but NP and AF.
Answer: We are grateful for your helpful advice. We have added some examples of whole IVD regeneration.
Whole tissue engineering IVD combines two approaches of NP replacement and AF repair together. The methods for constructing whole IVD can be divided into the following three categories: 1) cells-seeded scaffolds of NP and AF were prepared separately and assembled together into composite constructs. Nesti et al. used MSCs seeded PLLA electrospinning scaffold and HA gel and assembled them into an engineering IVD [1]. This composite scaffold provided a development of chondrocytic phenotype of the seeded cells. 2) Integrated biphasic NP-AF scaffolds. One research group developed an integrated biphasic NP-AF scaffolds from collagen and GAGs. A collagen-GAG co-precipitate core was comprised as the NP tissue and it is encapsulated in multiple lamellae of photo-chemically crosslinked collagen membranes, which comprise the AF-like lamellae [2]. This scaffold showed similar mechanical properties to native discs, with 82%-89% recovery of heights after mechanical loading, compared with a 99% recovery of native discs. 3) Scaffolds made of decellularized natural IVD. With chemistry and physics methods, Chan et al. made a 70% cells removing scaffolds in bovine IVDs [3]. This acellular scaffold maintains GAG content, structure of collagen fibers and biomechanical properties. Moreover, NP cells survive more than 7 days after implanted into the decellularized scaffold. (Page 8-9, lines 326-340, red font)
Nesti, L.J., et al., Intervertebral disc tissue engineering using a novel hyaluronic acid-nanofibrous scaffold (HANFS) amalgam. Tissue Engineering Part A, 2008. 14(9): p. 1527-1537.
Choy, A.T. and B.P. Chan, A Structurally and Functionally Biomimetic Biphasic Scaffold for Intervertebral Disc Tissue Engineering. PLoS One, 2015. 10(6): p. e0131827.
Chan, L.K.Y., et al., Decellularized bovine intervertebral disc as a natural scaffold for xenogenic cell studies. Acta Biomaterialia, 2013. 9(2): p. 5262-5272.
Requirements of ideal structure should be reported, in particular the need of cell proliferation control and angiogenesis
Answer: Thank you for your advice. We have added some contents of clinical requirements for tissue engineered scaffolds in IDD treatment.
With the development of tissue engineering, scaffolds satisfy the requirements of clinical treatment are regarded as the ‘holy grail’ to IDD repair. For NP repair: 1) Implanted biomaterials need to restore the height of IVD and the motion segment stability [1, 2]. 2) Implanted biomaterials should have sufficient durability, which means biomaterials can maintain physical support over millions of cycles of loading without generating minimal wear debris that may stimulate an immune response. 3) Implanted biomaterials should have a feasible environment for NP cell surviving and prevent the ingrowth of blood vessel because neovascularization facilitates the infiltration of macrophages into the IVD, triggering inflammation [3]. 4) Injectable materials is more appropriate because of its ability of causing minimal damage to the AF tissue. For AF repair: 1) Engineered AF scaffold that mimic the collagen fiber architecture of native tissue is the first choice in clinical trials. 2) Scaffolds reproduce the mechanical properties, strength and oriented microstructure of the native AF tissue are considered to be an ideal method for AF repair [4]. (Page 10, lines 370-381, red font)
Mehrkens, A., et al., Tissue engineering approaches to degenerative disc disease--a meta-analysis of controlled animal trials. Osteoarthritis Cartilage, 2012. 20(11): p. 1316-25.
Iatridis, J.C., et al., Role of biomechanics in intervertebral disc degeneration and regenerative therapies: what needs repairing in the disc and what are promising biomaterials for its repair? Spine J, 2013. 13(3): p. 243-62.
Walker, M.H. and D.G. Anderson, Molecular basis of intervertebral disc degeneration. Spine J, 2004. 4(6 Suppl): p. 158S-166S.
Bowles, R.D. and L.A. Setton, Biomaterials for intervertebral disc regeneration and repair. Biomaterials, 2017. 129: p. 54-67.
NP regeneration: Hydrogels are not biocompatible and with suitable mechanical properties per se’, but it depends on polymer/s used and fabrication methods, please revise;
Answer: Thank you for your comment. We have corrected it in the revised manuscript.
Profit from the improving fabricated techniques, hydrogel, one of the numerous biomaterials, has been pushed to the forefront of IVD treatment with many advantages such as high water content, excellent biocompatibility and three-dimensional (3D) network structure with biomechanical properties similar the natural IVD [1]. (Page 5, lines 163-166, red font)
Seliktar, D., Designing Cell-Compatible Hydrogels for Biomedical Applications. Science, 2012. 336(6085): p. 1124-1128.
Samples must be sterile always, that processing aspect is not just an advantage;
Answer: Thank you for your comment. We have edited it in the revised manuscript.
Chen et al, generated a high molecular weight hyaluronic acid-gelatin-adipic acid dihydrazide (oxi-HAG-ADH) hydrogel with several advantages including: 1) anti-inflammatory and immunosuppressive activities, necessary for clinical application; (Page 5, lines 174-176, red font)
Page 5 lines 163-171: it was reported in vitro evaluations without mentioning the cell type used;
Answer: We apologized about our carelessness. We have added the cell type in the revised manuscript.
bovine cells-seeded on SAPH, while total number of cells decreased with increase in load over time. Bovine cell-seeded in SAPH were less viscous and more elastic compared with the native NP. (Page 5, lines 191-193, red font)
AF regeneration: Just few examples were reported;
Answer: Thank you for your advice. We have added some AF regeneration examples in this part.
In tissue engineering strategies, natural materials such as collagen, hyaluronic acid (HA), chitosan, alginate, silk fibroin, and chondroitin sulfate (CS) are well established to AF regeneration. While some researchers preferred to design natural biologic materials such as decellularized matrix from AF to promote tissue regeneration and repair [1, 2]. Benefit from their origins, the natural scaffolds are endowed with advantages including low toxicity, similar properties to native tissue, and easy large-scale production. Synthetic polymers are obtained from industrial products and their mechanical and physicochemical properties can be finely adjusted. The most commonly synthetic materials used for AF scaffolds include poly (trimethylene carbonate) (PTMC), poly(lactide-co-glycolide) (PLGA), poly(ε-caprolactone) (PCL), poly(D, L-lactide) (PDLLA), poly(L-lactide) (PLLA), polyurethane, and HA-poly(ethylene glycol) (PEG) [3, 4]. These scaffolds can be fabricated and processed on the desired structure characteristics (aligned, angle-ply, hierarchical, bilayer, biphasic, etc) and mechanical properties of the final engineered tissue. (Page 6, lines 214-225, red font)
Xu, H., et al., Comparison of decellularization protocols for preparing a decellularized porcine annulus fibrosus scaffold. PLoS One, 2014. 9(1): p. e86723.
Bowles, R.D. and L.A. Setton, Biomaterials for intervertebral disc regeneration and repair. Biomaterials, 2017. 129: p. 54-67.
Nerurkar, N.L., et al., Dynamic culture enhances stem cell infiltration and modulates extracellular matrix production on aligned electrospun nanofibrous scaffolds. Acta Biomater, 2011. 7(2): p. 485-91.
Vadala, G., et al., Bioactive electrospun scaffold for annulus fibrosus repair and regeneration. Eur Spine J, 2012. 21 Suppl 1: p. S20-6.
Strategies in IVD displacement: just few examples were reported.
Answer: Thank you for your advice. We have added some examples in this part.
Besides these under developing biomaterials, a few of NP implants have been extensively used for IDD treatment. The prosthetic disc nucleus (PDN) is a hydrogel that can absorb up to 80% of its weight in water. PDN has passed FDA guidelines of cytotoxicity and biomechanical tests [116]. PDN can endure up to 50 million cycles with loads ranging from 200 N to 800 N. Aquarelle is made of a semihydrated poly vinyl alcohol (PVA) hydrogel. Animal tests showed that Aquarelle has a good biocompatibility and can tolerate up to 40 million cycles but high rates of extrusion was reported, ranging from 20%-30% depending on the approach [117]. (Page 7, lines 272-278, red font)
Ray, C.D., The PDN prosthetic disc-nucleus device. Eur Spine J, 2002. 11 Suppl 2: p. S137-42.
Allen, M.J., et al., Preclinical evaluation of a poly (vinyl alcohol) hydrogel implant as a replacement for the nucleus pulposus. Spine, 2004. 29(5): p. 515-523.
To date, artificial scaffolds are difficult to satisfy the requirements of AF engineering in clinical trials because the AF tissue has a complicated structure and unevenly distributed components. With the development of decellularization technique, many researchers have shifted their attentions to the decellularized tissue ECM. ECM scaffold regulates cell survival, proliferation and differentiation, moreover, it is an ideal carrier for growth factors and cytokines attaching and delivering in vivo. One research group developed a decellularized porcine AF scaffold by using chemical reagents and biological enzyme to remove pig AF cells. With biological and mechanical tests, results showed that the decellularized porcine AF scaffold maintained the similar structure and components compared to the native AF tissue. Mechanical property showed no significant difference between the scaffold and native AF. Most important, rabbit AF cells seeded into the scaffold showed good viability implying the scaffold possessed favorable biocompatibility [1]. (Page 8, lines 314-325, red font)
Xu, H., et al., Comparison of decellularization protocols for preparing a decellularized porcine annulus fibrosus scaffold. PLoS One, 2014. 9(1): p. e86723.
Whole tissue engineering IVD combines two approaches of NP replacement and AF repair together. The methods for constructing whole IVD can be divided into the following three categories: 1) cells-seeded scaffolds of NP and AF were prepared separately and assembled together into composite constructs. Nesti et al. used MSCs seeded PLLA electrospinning scaffold and HA gel and assembled them into an engineering IVD [1]. This composite scaffold provided a development of chondrocytic phenotype of the seeded cells. 2) Integrated biphasic NP-AF scaffolds. One research group developed an integrated biphasic NP-AF scaffolds from collagen and GAGs. A collagen-GAG co-precipitate core was comprised as the NP tissue and it is encapsulated in multiple lamellae of photo-chemically crosslinked collagen membranes, which comprise the AF-like lamellae [2]. This scaffold showed similar mechanical properties to native discs, with 82%-89% recovery of heights after mechanical loading, compared with a 99% recovery of native discs. 3) Scaffolds made of decellularized natural IVD. With chemistry and physics methods, Chan et al. made a 70% cells removing scaffolds in bovine IVDs [3]. This acellular scaffold maintains GAG content, structure of collagen fibers and biomechanical properties. Moreover, NP cells survive more than 7 days after implanted into the decellularized scaffold. (Page 8-9, lines 326-340, red font)
Nesti, L.J., et al., Intervertebral disc tissue engineering using a novel hyaluronic acid-nanofibrous scaffold (HANFS) amalgam. Tissue Engineering Part A, 2008. 14(9): p. 1527-1537.
Choy, A.T. and B.P. Chan, A Structurally and Functionally Biomimetic Biphasic Scaffold for Intervertebral Disc Tissue Engineering. PLoS One, 2015. 10(6): p. e0131827.
Chan, L.K.Y., et al., Decellularized bovine intervertebral disc as a natural scaffold for xenogenic cell studies. Acta Biomaterialia, 2013. 9(2): p. 5262-5272.

Reviewer 5 Report
The review article is well-written. It introduces the role of disordered mechanical stress in interveterbral disc degeneration and treatment strategy. I suggest the publication of the work.
Author Response
Replies to Reviewer 5
Comments and Suggestions for Authors
The review article is well-written. It introduces the role of disordered mechanical stress in interveterbral disc degeneration and treatment strategy. I suggest the publication of the work.
Answer: Thank you for your positive commet.

Round 2
Reviewer 1 Report
The revised version is much improved. If I could, I might suggest adding a table summarizing tissue engineering / hydrogel attempts to date, (NP vs AF vs whole IVD, test species, length of test, etc).
Author Response
Comments and Suggestions for Authors
The revised version is much improved. If I could, I might suggest adding a table summarizing tissue engineering / hydrogel attempts to date, (NP vs AF vs whole IVD, test species, length of test, etc).
Answer: Thank you for your kind advice. We have added a table to summarize the attempts of tissue engineering in IDD.
There are numerous studies that chronicle the development and application of tissue engineering based therapies in IDD (Table 2). Although these studies are versatile, they can be divided into two areas as follows: regenerative therapies and displacement therapies in IVD. (Page 4, lines 144 -146, red font)
Table 2 Summary of tissue engineering strategies in IDD treatment
Tissue engineering strategies in NP treatment | |||
Materials | Test species | Test time | Results |
PLGA | Dog | 8-week | PLGA with cells significantly maintained the height and the stability of disc [1]. |
Fibrin | Pig | 12-week | Fibrin significantly inhibited the fibrosis and inflammation of NP and enhanced the synthesis of ECM [2]. |
Collagen II (CII)/hyaluronate (HyA)/chondroitin-6-sulfate (6-CS) | Rabbit | 84-day | The CII/HyA-CS scaffolds have a highly porous structure, high water-binding capacity and significantly improved mechanical stability. This scaffolds also showed satisfactory biocompatibility [3]. |
PGA-hyaluronan | Rabbit/Sheep | 12 month/6 month | Enhanced repair tissue formation and MRI intensity [4, 5] |
Silk fibroin (SK) /polyurethane (PU) composite | Pig | NA | SK/PU is an injectable hydrogel with minimally invasive treatment, suitable physical-mechanical properties, and visible CT and T2-weight MRI [6]. |
Modified hyaluronic acid gels | Pig | 6-week | Both HYAFF® 120 and HYADD 3® treatment supported an NP-like region forming and prevented IVD narrowing, fibrous tissue replacement and bony end-plates disruption [7]. |
Tissue engineering strategies in NP treatment in AF treatment | |||
Electrospun PCL | Rat | 4-week | PCL can mimick the hierarchical organization of the native AF and achieve functional partly with native tissue [8]. |
Photochemically crosslinked collagen in shape of needle | Rabbit | 1 month | Materials can sustain the physiologically relevant loadings, prevent leakage and reduce osteophyte formation [9]. |
Collagen-fibrin gel scaffolds | Rabbit cells in vitro | 4 months | Collagen-fibrin gel significantly delayed the fibrous tissue infiltration. GAG and hydroxyproline content increase over four months [10]. |
Tissue engineering strategies in the whole IVD | |||
AF-polyglycolic acid and polylactic acid NP-alginate | Mice | 12-week | The engineered disc maintained the gross morphology and the AF was rich type I collagen but NP contained type II collagen [11]. |
AF-contracted collagen, NP-alginate | Rat | 6 months | Tissue-engineered IVD maintained disc space height, produced de novo extracellular matrix, and integrated into the spine, yielding an intact motion segment with dynamic mechanical properties similar to that of native IVD [12]. |
AF- poly (butylene succinate-co-terephthalate) copolyester (PBST), NP-chitosan hydrogel | Rabbit | 4-week | The whole TE-IVD stimulated the natural structure of IVD and retained the height of IVD after four weeks of implant [13]. |
1. Ruan, D.K., et al., Experimental intervertebral disc regeneration with tissue-engineered composite in a canine model. Tissue Eng Part A, 2010. 16(7): p. 2381-9.
2. Buser, Z., et al., Biological and biomechanical effects of fibrin injection into porcine intervertebral discs. Spine (Phila Pa 1976), 2011. 36(18): p. E1201-9.
3. Li, C.Q., et al., Construction of collagen II/hyaluronate/chondroitin-6-sulfate tri-copolymer scaffold for nucleus pulposus tissue engineering and preliminary analysis of its physico-chemical properties and biocompatibility. J Mater Sci Mater Med, 2010. 21(2): p. 741-51.
4. Endres, M., et al., Intervertebral disc regeneration after implantation of a cell-free bioresorbable implant in a rabbit disc degeneration model. Biomaterials, 2010. 31(22): p. 5836-41.
5. Woiciechowsky, C., et al., Regeneration of nucleus pulposus tissue in an ovine intervertebral disc degeneration model by cell-free resorbable polymer scaffolds. J Tissue Eng Regen Med, 2014. 8(10): p. 811-20.
6. Hu, J., et al., Injectable silk fibroin/polyurethane composite hydrogel for nucleus pulposus replacement. J Mater Sci Mater Med, 2012. 23(3): p. 711-22.
7. Revell, P.A., et al., Tissue engineered intervertebral disc repair in the pig using injectable polymers. J Mater Sci Mater Med, 2007. 18(2): p. 303-8.
8. Martin, J.T., et al., Translation of an engineered nanofibrous disc-like angle-ply structure for intervertebral disc replacement in a small animal model. Acta Biomater, 2014. 10(6): p. 2473-81.
9. Chik, T.K., et al., Photochemically crosslinked collagen annulus plug: a potential solution solving the leakage problem of cell-based therapies for disc degeneration. Acta Biomater, 2013. 9(9): p. 8128-39.
10. Pan, Y., et al., Cells scaffold complex for Intervertebral disc Anulus Fibrosus tissue engineering: in vitro culture and product analysis. Mol Biol Rep, 2012. 39(9): p. 8581-94.
11. Mizuno, H., et al., Tissue-engineered composites of anulus fibrosus and nucleus pulposus for intervertebral disc replacement. Spine (Phila Pa 1976), 2004. 29(12): p. 1290-7; discussion 1297-8.
12. Bowles, R.D., et al., Tissue-engineered intervertebral discs produce new matrix, maintain disc height, and restore biomechanical function to the rodent spine. Proc Natl Acad Sci U S A, 2011. 108(32): p. 13106-11.
13. Yuan, D., et al., The establishment and biological assessment of a whole tissue-engineered intervertebral disc with PBST fibers and a chitosan hydrogel in vitro and in vivo. J Biomed Mater Res B Appl Biomater, 2019.

Reviewer 3 Report
Accept after minor English grammar corrections.
Author Response
Comments and Suggestions for Authors
Accept after minor English grammar corrections.
Answer: Thank you for your positive comment and kind advice. We have correct some grammar mistakes in the review.
1. These support a new understanding that disordered mechanical stress plays an important role in the degeneration of the intervertebral discs. (Page 1, lines 16-17, red font)
2. In this paper, we will review the current literatures available on the role of disordered (Page 1, lines 19, red font)
3. Intervertebral disc degeneration (IDD) is a chronic disease that which slowly degrades the content of intervertebral disc (IVD) (Page 1, lines 29-30, red font)
4. However, these strategies are restricted to relieving the pain and symptoms without eliminating the disease itself due to an incomplete understanding of the pathobiology patho-biology of IDD (Page 1, lines 33-34, red font)
5. Midsagittal cross-section showing anatomical regions (Page 2, lines 47-48, red font)
6. The altered biomechanics in IVD has have been widely accepted as an important contributor to IDD because biomechanical loading directly affects IVD cell metabolism (Page 2, lines 49-50, red font)
7. NP stems from the axial notochord. (Page 2, line 54, red font)
8. Taken together, disordered mechanical stress may become another reason for IDD besides biochemistry except for the common concept of biochemistry factor. (Page 2, lines 62-64, red font)
9. Strategies for of restoring the structures of NP and AF are essential in the clinical trials. (Page 2, lines 66-67, red font)
10. Various tissue-engineered scaffolds with feasible substances (e.g. functional cells, growth factors, proteins, peptides, etc.) (Page 2, lines 70-71, red font)
11. We expect that this review will support a new direction to an understanding of IDD mechanism and therapy application. (Page 2, lines 73-74, red font)
12. NCs are the main cell type of the NP during the embryonic period. (Page 2, lines 80-81, red font)
13. and protects CLCs against apoptosis (Page 3, line 88, red font)
14. Taking into account that humans commonly lose lost NCs in adolescence adolescents and the cost of experimental animals (such as pigs and dogs), rodents are the favored animal model to investigate the role of NCs in IDD (Page 3, lines 93-95, red font)
15. Depending on the direction of the inner AF and the outer AF, the thickness of the collagen fiber lamellae varies vary between 200 and 400μm (Page 3, lines 108-110, red font)
16. Collagen alignment is the key to the functional role of the AF (Page 3, line 113, red font)
17. Mechanical stress influences the AF cells through several cytoskeletal (Page 3, line 118, red font)
18. Studies have shown that disordered mechanical stress may expose the cleavage binding sites by changing in the spatial structure of collagen monomers and three alpha chains (Page 4, lines 124-126, red font)
19. Accumulating studies demonstrated that angiogenesis was existed in IDD. Disordered mechanical stress facilitates the neovascularization through by destroying the physical barriers (Page 4, lines 131-132, red font)
20. Moreover, one study demonstrated that disordered mechanical stress can directly influence the ingrowth of blood vessels (Page 4, lines 133-134, red font)
21. Human AF cells experienced cyclic tensile strain showed a nearly 70% increase of gene expression of pleiotrophin, in which the pleiotrophin is regarded as a pathologic alteration of disc tissue and its neovascularization (Page 4, lines 134-137, red font)
22. Disordered mechanical stress induces NCs apoptosis which promotes the proliferation of CLCs and maintains the normal function of NP (Page 4, lines 141-142, red font)
23. and to the ingrowth of blood vessels further impair the integrity of IVD (Page 4, line 144, red font)
24. Disordered mechanical stress causes IVD degeneration. (Page 4, line 149, red font)
25. the periphery to center, and infiltration of AF lamellae into the NP and loss of the annular-nuclear demarcation. In late stage, the mechanical disorder causes serious degeneration, clefts extend through nucleus and annulus, and the EPs change into diffused sclerosis (Page 5, line 153-155, red font)
26. high water content that allows it to transmit stress and resist compressive forces when When unloaded (Page 6, line 163, red font)
27. In recent years, bioengineered scaffolds that are similar to the native NP structure and mechanical properties property have been gaining attention. (Page 6, lines 165-166, red font)
28. Apart from being cost effective, hydrogels fabricated with high-molecular-weight polymers (Page 7, line 178, red font)
29. With a better understanding of IDD mechanism, functions, as well as NP-relate proteins have been discovered, (Page 7, lines 185-186, red font)
30. These hydrogels provided a favorable environment for NP cell, proliferation and more importantly, promoted the expression of specific markers characteristic of immature NP cells (Page 7, lines 191-192, red font)
31. Besides laminin derived peptides (Page 7, line 194, red font)
32. while the total number of cells decreased with increase in load over time. (Page 7, lines 198-199, red font)
33. Bovine cell-seeded in SAPH was were less viscous and more elastic compared with the native NP. (Page 7, lines 199-200, red font)
34. have contributed to a better understanding of molecular mechanisms and cellular pathways in several human diseases including IDD. (Page 7, lines 213-215, red font)
35. functional AF regeneration with an increase in collagen fibril formation in Mkx-/- mice. (Page 8, lines 220-221, red font)
36. For suppressing the ingrowth of vessels, normally tissue engineering strategies focus on rebuilding the construction of IVD. (Page 8, lines 244-245, red font)
37. Therefore, some anti-angiogenesis hydrogels were developed to against the neovascularization in IDD. (Page 8, lines 249-250, red font)
38. Cell study demonstrated that endothelial cells could not adhere to the gel surface and endothelial cells showed a significant lower viability compared with cells seeded on matrigel. (Page 8, lines 251-253, red font)
39. A Ssimilar study used iGG-MA hydrogel containing a VEGF blocker peptidic aptamers sequence (WHLPFKC), results showed that the functional hydrogel not only prevented vessel ingrowth, but also induce their regression at the tissue/iGG-MA interface (Page 8, lines 258-260, red font)
40. Taking into account the requirements of the clinic, (Page 8, line 264, red font)
41. Third, minimal/non-invasive strategies to deliver injectable materials without causing further damages to the already degenerating IVD are necessary in for clinic. (Page 8, lines 266-268, red font)
42. Animal tests showed that Aquarelle has a good biocompatibility and can tolerate up to 40 million cycles but high rates of extrusion were was reported, ranging from 20%-30% depending on the approach (Page 9, lines 284-286, red font)
43. One research group developed a decellularized porcine AF scaffold by using chemical reagents and biological enzymes to remove pig AF cells. (Page 10, lines 327-329, red font)
44. The Mmechanical property showed no significant difference between the scaffold and native AF. (Page 10, lines 331-332, red font)
45. This acellular scaffold maintains GAG content, the structure of collagen fibers and biomechanical properties. Moreover, NP cells survive more than 7 days after implanted into the decellularized scaffold. (Page 11, lines 346-348, red font)
46. It is well known that the mechanical environment can affect cellular homeostasis [144]. NP cells with a decline in osmotic pressure exhibit a decreased synthesis of aggrecan and an increased production of MMP-3 which initiate NP degeneration (Page 11, lines 352-355, red font)
47. With the degeneration of the NP, an increasing shear stress and a decreasing swelling pressure will lead to a formation of a fibrous tissue because of the depositing of collagen type I (Page 11, lines 355-357, red font)
48. Besides paying attention to the NP/AF repair, some studies have focused on the other factors induced by IDD (Page 11, lines 361-362, red font)
49. EFTs could promoted NP cells against apoptosis and suppress the process of IDD (Page 11, line 367, red font)
50. For example, in NP regeneration, growth factor (GF) is are only sufficient to the resident cells that still response to GF treatment. (Page 11, lines 369-370, red font)
51. thus, the genetic modified cell-seeded scaffold may be employed to enhance the synthesis of GFs and allow sustained secretion of anabolic proteins (Page 11, lines 372-373, red font)
52. Implanted biomaterials should have a feasible environment for NP cells surviving and prevent the ingrowth of blood vessels (Page 12, lines 384-383, red font)
53. Injectable materials are is more appropriate because of its ability to cause of causing minimal damage to the AF tissue. (Page 12, lines 385-386, red font)
54. More advanced materials are required in clinically strategies. (Page 12, lines 394-395, red font)
Reviewer 4 Report
The manuscript was largely revised by following referees comments-suggestions, as a consequence the scientific impact was improved.
Just one comment:
page 5 line 203, hydrogels are not bocompatible per se', even if can provide an aqueous environment to cells, but it depends on chemistry. Please revise.
Author Response
Comments and Suggestions for Authors
The manuscript was largely revised by following referes comments-suggestions, as a consequence the scientific impact was improved.
Just one comment:
page 5 line 203, hydrogels are not biocompatible per se', even if can provide an aqueous environment to cells, but it depends on chemistry. Please revise.
Answer: Thank you for your kind advice. We have revised this part in the review.
Hydrogel is not biocompatible and does not have suitable mechanical properties per se'. However, by changing the polymer type and optimizing the fabrication methods, hydrogel could obtain high water content, good biocompatibility, 3D network structure and suitable biomechanical property which similar to the natural IVD (Page 6, lines 167 -171, red font)